# Challenges of an Emerging Disease: The Evolving Approach to Diagnosing Devil Facial Tumour Disease

**DOI:** 10.3390/pathogens11010027

**Published:** 2021-12-28

**Authors:** Camila Espejo, Amanda L. Patchett, Richard Wilson, A. Bruce Lyons, Gregory M. Woods

**Affiliations:** 1Tasmanian School of Medicine, College of Health and Medicine, University of Tasmania, Hobart, TAS 7000, Australia; camila.espejo@utas.edu.au (C.E.); bruce.lyons@utas.edu.au (A.B.L.); 2Menzies Institute for Medical Research, University of Tasmania, Hobart, TAS 7000, Australia; amanda.patchett@utas.edu.au; 3Central Science Laboratory, University of Tasmania, Hobart, TAS 7000, Australia; richard.wilson@utas.edu.au

**Keywords:** DFTD, laboratory diagnosis, transmissible cancer, histopathology, proteomics, bioinformatics, extracellular vesicles

## Abstract

Devil Facial Tumour Disease (DFTD) is an emerging infectious disease that provides an excellent example of how diagnostic techniques improve as disease-specific knowledge is generated. DFTD manifests as tumour masses on the faces of Tasmanian devils, first noticed in 1996. As DFTD became more prevalent among devils, karyotyping of the lesions and their devil hosts demonstrated that DFTD was a transmissible cancer. The subsequent routine diagnosis relied on microscopy and histology to characterise the facial lesions as cancer cells. Combined with immunohistochemistry, these techniques characterised the devil facial tumours as sarcomas of neuroectodermal origin. More sophisticated molecular methods identified the origin of DFTD as a Schwann cell, leading to the Schwann cell-specific protein periaxin to discriminate DFTD from other facial lesions. After the discovery of a second facial cancer (DFT2), cytogenetics and the absence of periaxin expression confirmed the independence of the new cancer from DFT1 (the original DFTD). Molecular studies of the two DFTDs led to the development of a PCR assay to differentially diagnose the cancers. Proteomics and transcriptomic studies identified different cell phenotypes among the two DFTD cell lines. Phenotypic differences were also reflected in proteomics studies of extracellular vesicles (EVs), which yielded an early diagnostic marker that could detect DFTD in its latent stage from serum samples. A mesenchymal marker was also identified that could serve as a serum-based differential diagnostic. The emergence of two transmissible cancers in one species has provided an ideal opportunity to better understand transmissible cancers, demonstrating how fundamental research can be translated into applicable and routine diagnostic techniques.

## 1. Introduction

During their training, medical students are taught the importance of extracting an accurate patient’s clinical history. The history will reveal valuable clues to a skilled medical practitioner that will inform a potential diagnosis, and the appropriate laboratory tests will confirm the diagnosis. This process is particularly relevant to diagnosing infectious diseases and cancer, as a prompt and accurate diagnosis relies on the correct laboratory test. Veterinary practitioners do not have the luxury of questioning their “patient” and rely on observational skills. However, confirmational diagnosis often relies on the results from multiple laboratory tests.

## 2. Diagnosis of Infectious Diseases and Cancer

The laboratory identification of infectious diseases and cancer can range from basic to complex procedures. For infectious diseases, microscopy is relatively basic and can be used to directly visualise a stained specimen (e.g., blood, sputum and urine) for the presence of microorganisms. More complex procedures include the extraction of genetic material, which can be analysed by molecular techniques such as polymerase chain reaction (PCR) and sequencing. PCR requires knowledge of the target organism, whereas sequencing DNA involves bioinformatic analyses but does not require prior knowledge of the microorganism. Less complex procedures include serology, which can indirectly determine the presence of microorganisms based on the antibodies produced by the host in response to the infection.

Physical examinations, familial cancer history and imaging techniques (e.g., X-ray, CT and MRI) are often the initial procedures used when diagnosing cancer in humans. Laboratory identification and/or confirmation usually requires a biopsy that can be stained in a pathology laboratory. An experienced pathologist will use microscopy to identify the nature of the cancer cells. Biomarkers primarily in the blood (e.g., IgG for myeloma) but occasionally in the urine (e.g., Beta-2 microglobulin for leukaemia) or in the cerebrospinal fluid (e.g., fibrin for bladder cancer) can assist in diagnosis and monitoring. Molecular tests such as PCR, chromosomal analysis, fluorescent in situ hybridisation (FISH) and genetic sequencing provide additional diagnostic data. It should be noted that no single test is sufficient for a reliable diagnosis. An accurate cancer diagnosis is often a complex procedure that involves a range of different diagnostic techniques.

Cancer can be suspected in domestic animals based on the owner providing clues (e.g., behavioural changes, presence of lumps and loss of appetite) and a veterinarian providing a physical examination. Cancer diagnosis in pets is like a human cancer diagnosis, but it is usually not as elaborate due to economic constraints. The most common are blood tests to identify abnormal cell counts and paraneoplastic syndrome, fine needle aspirates and biopsies to look for abnormal cells and imaging techniques. Advances in cancer genomics have provided a greater understanding of cancer. Initially, a research tool, genomics, identified “one health” similarities between humans and dogs and is gradually being implemented into the cancer diagnosis of dogs [1]. Cancer diagnosis, pathology and treatment in the area of “one health” can benefit humans and dogs [2], as animals and human cancers share similar features, including genomic and environmental predisposition factors. Similar studies of cancer of wild animals are not usually afforded such luxuries, except when interesting cases arise, such as transmissible cancers in dogs and Tasmanian devils.

## 3. Devil Facial Tumour Disease

In 1996, the presence of lesions on the faces of several devils located at wukalina/Mount William National Park in Tasmania’s northeast was noted by a wildlife photographer. Concerns increased as more devils from different areas of Tasmania showed signs of massive tumours, and the devil abundance appeared to be in dramatic decline [3]. It became apparent that the tumours were more than an ordinary mass of cells. Standard histopathology and electron microscopy were performed on tumour biopsies [4]. Macroscopically, the tumours were soft tissue masses that often ulcerated. The cells were round to spindle-shaped (Figure 1A), and electron microscopy could only identify a few specialised organelles such as secretory granules, desmosome-like structures and myelin bodies. This was not enough evidence to classify DFTD as a carcinoma but could exclude tumours such as lymphomas. There was no evidence of viral particles, ruling out a viral infection [4]. A preliminary classification of the tumours, which looked similar from all animals examined, was “undifferentiated soft tissue neoplasm”. This fatal emerging neoplasm was named Devil Facial Tumour Disease (DFTD) after the clinical signs of tumours on the facial, oral and neck regions. DFTD spread rapidly, affecting more than 51% of the devil population by mid-2005 [3].

Immunohistochemistry, another relatively basic technique that relies on antibody binding to detect specific cells, was used to further diagnose the nature of these unusual tumours. There were no reagents specific for Tasmanian devils available commercially; therefore, a range of antibodies was screened for cross-reactivity with Tasmanian devil tissues. Fortunately, there was sufficient cross-reactivity to produce a refined diagnosis of “undifferentiated, sub-epithelial sarcoma of neuroectodermal origin” [5]. This diagnosis was refined with sophisticated analyses of messenger ribonucleic acid (mRNA). Deep sequencing of the DFTD transcriptome and microRNAs identified DFTD as a cancer of a Schwann cell origin [6].

Laboratory techniques, such as immunohistochemistry of suspected cancer tissue, led to an accurate identification of DFTD. Of the multiple neuroectodermal markers identified in DFTD, we found that periaxin was consistently expressed by DFTD cells [7]. Although periaxin was also expressed by peripheral nerve cells, DFTD cells could be distinguished from these nerves due to the peripheral nerves’ distinctive parallel fibre arrangement (Figure 1B) [7]. The use of periaxin exemplified the translation of research finding into an excellent analytical tool for the routine diagnosis of DFTD. However, there still remains a scope for the further development of more rapid and inexpensive diagnostic tools.

Chromosomal changes can be associated with many cancers, and a basic karyotype can detect relatively gross changes (e.g., chromosomal abnormalities in leukemic cells). Given the specialised nature of this technique and application for limited diseases, karyotyping samples from animals is rare. One notable exception is DFTD, as karyotyping provided the initial evidence that DFTD was a transmissible cancer [8]. Confirmation of the transmissibility and clonal origin of DFTD came from more sophisticated genetic techniques using microsatellites, major histocompatibility complex (MHC) markers and genome analyses. Siddle et al. [9] found that 15 separate tumour samples showed identical microsatellite and MHC loci, supporting the clonal nature of DFTD. A subsequent study demonstrated that 25 tumour samples had an identical genotype, independent of the animal’s location, sex or age [10]. An analysis of the whole genome also confirmed the allograft theory of transmission, as the DFTD tumours were genotypically different from their host and shared multiple genetic markers [10,11].

An extension of karyotyping is “chromosome painting”, whereby the DNA on individual chromosomes can be fluorescently labelled, each chromosome a different colour. Chromosome painting is particularly useful for identifying chromosomal rearrangements unique to specific cancers, DFTD being one example. The chromosome painting on DFTD tumour strains determined the origins of marker chromosomes and rearrangements of DFTD karyotypes [12]. Using similar techniques, Murchison, et al. [10] proposed that DFTD arose from a female devil, as there were complex rearrangements of two X chromosomes.

In 2016, we published the description of a second transmissible DFTD cancer [13]. With characteristic DFTD gross morphology and histology features, it was justifiably assumed to be DFTD. However, immunohistochemistry alerted to something different, as the cancer cells did not express periaxin (Figure 1C,D). Furthermore, the karyotype of this cancer indicated a newly arrived and independent facial cancer, now referred to as DFT2 (Figure 2) [13]. For the remainder of this review, the first DFTD will be referred to as DFT1 and the second DFTD as DFT2.

DFT2 tumours also presented evidence of allograft transmission and clonality, being karyotypically and genetically different from both DFT1 tumours and their host. DFT2 karyotypes exhibited identical complex structural abnormalities (Figure 2) [13]. While the DFT1 karyotype lacks intact sex chromosomes, DFT2 revealed the existence of X and Y chromosomes, indicating a male origin [13]. Genetic analyses confirmed that DFT2 presented a different genotype to both its host and DFT1 at microsatellite and MHC loci, reaffirming the allograft nature of DFT2 [13].

All DFTD tumours are tentatively diagnosed by the appearance of macroscopic tumours, but the differentiation between DFT1 and DFT2 provides a diagnostic challenge. Once a tumour biopsy or a fine needle aspiration sample is collected, multiple complementary laboratory techniques are required to confirm DFTD and to differentiate between DFT1 and DFT2. Histopathology demonstrated that DFT1 tumours are composed of pleomorphic round cells with a high central nucleus and indistinct cell borders, arranged in bundles or nests and surrounded by a thin fibrous pseudocapsule [4]. In contrast, DFT2 tumour cells are characterised by sheets of pleomorphic (amorphic to stellate and fusiform) cells distributed in solid patterns [13]. A distinguishing histological feature was the absence of detectable periaxin expression in DFT2. Even though immunohistochemistry techniques accurately confirmed DFTD tumours, it was challenging to distinguish DFT1 from DFT2. As cytogenetics (karyotyping) confirmed the clonal nature of DFT1 [8], it was used to corroborate the presence of DFT1 and/or DFT2. Although accurate, cytogenetics is time-consuming and requires expertise in collecting the sample, especially fine needle aspirates. Factors that can limit the success of in vitro DFTD cultures include fibroblast contamination, variations in temperature during transport to the laboratory and microbial contamination from the sample site [14]. For these reasons, a simple PCR-based diagnostic assay (Tasman-PCR) was developed to identify and distinguish DFT1 and DFT2 by amplification of a single polymorphism among DFT1, DFT2 and the host DNA [15]. Although Tasman-PCR is a rapid and highly sensitive and specific technique, it has limitations. For example, potential DNA cross-contamination can produce false-positive results. Sensitivity is limited for samples with very low tumour cell abundance that usually come from fine needle aspirations of small nonulcerated tumours, as biopsies from them are avoided (especially inside the oral cavity) [15]. Thus, the likelihood of not obtaining any DFT1/DFT2 tumour cells is high when collecting fine needle aspiration samples from small and nonulcerated tumours.

## 4. Differentiation of DFTDs (DFT1 and DFT2)

The unprecedented discovery of two transmissible cancers in Tasmanian devils prompted studies into their genesis. DFT1 and DFT2 were originally classified as neuroectodermal cancers due to their high expression of tissue markers such as vimentin, neural-specific enolase and S100 [5,16]. Further studies led by Murchison et al. [6] and Patchett et al. [17] sought to determine the specific cell of origin of DFT1 and DFT2 using transcriptomic and proteomic approaches. By comparing the tumours with a range of healthy Tasmanian devil tissues such as spleen, heart, brain and testes, it was revealed that both DFT1 and DFT2 were transcriptionally most similar to peripheral nerve tissue [6,17]. Furthermore, both cancers expressed lineage markers associated with Schwann cells, the myelinating cell of the peripheral nerve, including SOX10, nestin and nerve growth factor receptor [6,17]. These findings suggested a similar Schwann cell origin of both the DFT1 and DFT2 cancers.

The identification of DFT2 as a Schwann cell cancer was initially surprising due to the low expression of myelin-specific proteins such as periaxin by this cancer [13]. However, our subsequent analysis revealed that this expression pattern was reminiscent of both immature Schwann cells and non-myelinating Schwann cells, with roles in nerve maintenance [17,18]. Indeed, DFT2 cancers have an enriched gene profile similar to a ‘repair’ Schwann cell, a specialised mesenchymal Schwann cell involved in tissue and peripheral nerve repair during injury [19,20,21]. Repair Schwann cells develop from mature Schwann cells via an epithelial-to-mesenchymal-like transition prompted by cytokines released at the site of tissue injury [21,22]. It has been hypothesised that DFT2 cells, which express MHC-I molecules [23], could be under greater immune pressure in the tumour microenvironment, giving rise to this immune-induced repair Schwann cell phenotype [17,18]. In contrast, MHC-I loss in DFT1 [24] has likely reduced the immune pressure these cancers face, thus promoting the typical myelinating phenotype of Schwann cells observed in DFT1.

The contrasting phenotypic states of DFT1 and DFT2 could potentially be harnessed to distinguish the cancers. We proposed that detection of the Schwann cell lineage marker SOX10 be included in immunohistochemistry protocols with periaxin detection to diagnose and distinguish DFT1 and DFT2 [17]. Alternatively, proteomic or transcriptomic approaches could be used to detect a panel of differentially expressed genes and proteins that distinguish DFT1 and DFT2 (Figure 3A). The challenge of this approach is that it relies on DFT1 and DFT2 phenotypes remaining stable throughout adaptive tumour evolution. This could be an unlikely characteristic of DFT cancers, given the highly plastic nature of Schwann cells during peripheral nerve injury [19,20]. Indeed, our recent findings suggested that DFT1 cancers can alter their phenotype during changing immune conditions, transiting into a mesenchymal phenotype with similarities to DFT2 when under immune pressure [25]. As in human cancers, this increase in mesenchymal activation likely enhances tumour growth, immune evasion, invasion and migration to promote continued oncogenesis [26]. Importantly, this finding suggests that the respective differentiation states of DFT1 and DFT2 can exhibit plasticity under different conditions (Figure 3B). This finding should be considered, and caution should be employed when gene or protein markers associated with the differentiation of DFT1 and DFT2 are used to distinguish the two cancers.

## 5. Liquid Biopsy to Differentiate DFTDs (DFT1 and DFT2)

The gold standard method to diagnose different types of cancers is based on solid (tumour) biopsies, which can be highly invasive and potentially increase the risks of haemorrhage and metastasis, hampering serial monitoring [28,29]. Furthermore, tumours are not always accessible for a tumour biopsy [28]. An alternative to solid tumour biopsies is liquid biopsies, which detect cancer-derived products and markers that reflect a cancer’s bodily fluids status [30]. Compared to traditional tumour biopsies, liquid biopsies are minimally invasive, as they only require routinely collected body fluids and are always available, independent of the location of the tumour [31]. Moreover, liquid biopsies have the advantage of serial monitoring, providing information related to disease progression or response to treatments [31]. Liquid biopsy analytical targets include circulating tumour cells, circulating tumour DNA, tumour-educated platelets and extracellular vesicles (EVs) [32]. EVs are nanosized bilipid structures released by most cells, including cancer cells, to their extracellular environment [33]. EVs are critical players in intercellular communication processes, including mechanisms of cancer progression via their bioactive cargo, such as proteins, lipids and nucleic acids [34,35]. The analysis of EV cargo has identified cancer biomarkers, including those that can differentiate between cancer subtypes [36,37], positioning EVs as a promising target for differential diagnostic liquid biopsies.

Considering the promise that EVs offer to discover cancer biomarkers, they have recently been explored in the DFTD system. In our experience, we found that cultured DFT2 cells released EVs in significantly greater numbers than cultured DFT1 cells [38]. These DFT2-released EVs were enriched in mesenchymal proteins, with a greater nanoparticle abundance in the serum of DFT2-infected devils relative to those infected with DFT1. Our study reported the discovery of an EV protein as a potential DFT2 biomarker. The extracellular matrix glycoprotein tenascin C (TNC), a mesenchymal protein enriched in cultured DFT2 cell-derived EVs, had a high predictive value to classify devils with DFT2 from those infected with DFT1 (100% sensitivity and 91.7% specificity). Moreover, TNC distinguished devils with DFT2 from healthy controls. These results demonstrated that TNC is a potential EV biomarker candidate to distinguish DFTD tumours, and as it can be found in peripheral blood serum, it may enable a liquid biopsy for DFT2 when tumours cannot be sampled.

## 6. Pre-Diagnostic Approaches

Blood tests have been routinely used to assist in the diagnosis of cancer. An example from human medicine is the early detection of pancreatic cancer by detecting the CA19-9 biomarkers in fluids, such as blood [39]. Prostate-specific antigen is another commonly utilised biomarker found in blood and can be used as an early detection test [40]. Circulating cancer cells can be detected in the blood of patients with early signs of cancer. Often, the cancer cells need to be enriched from a blood sample to provide enough detectable material for detection by various techniques, of which RT-PCR is the most common for the detection of cancer cell mRNAs [41]. All the diagnostic techniques currently used in the laboratory for DFTD require the collection of a tumour biopsy, which is not always possible, as it is limited to ulcerated tumours [14]. Biopsies are not possible when tumours are nonulcerated or not evident at the clinical examination (latent DFTD). There is direct evidence that DFT1 can have a long latent period before tumours are visible, as wild devils brought into isolated captivity have developed DFT1 up to 13 months after removal from the wild [42]. Two studies have investigated whether metabolites or proteins found in devil serum samples could serve as preclinical biomarkers to detect DFT1 during the latent period. One of these studies demonstrated that a panel of fibrinogen peptides and seven metabolites could distinguish DFT1-infected devils from healthy controls with high sensitivity and specificity [43]. The other study reported elevated levels of the receptor tyrosine–protein kinase ERBB3 in the serum of devils infected with DFT1 relative to the healthy controls [44]. As these two studies demonstrated the feasibility of finding potential DFT1 biomarkers in raw serum, more sensitive approaches based on EVs will be discussed to detect DFTD early.

## 7. Extracellular Vesicles

One of the main advantages of serum tests is the potential to detect cancer at the early stages [45]. The analysis of EV-associated proteins has yielded several early human cancer biomarkers that could be detected in serum [46,47,48,49]. The investigation of EVs related to DFTD has demonstrated the potential of this approach for early diagnosis. Since the emergence of a coordinated response to the DFT1 epidemic, developing a test for the early detection of DFTD has been a high priority for conservation management [50]. Using contemporary methodology for the isolation of EVs and quantitative proteomics, we demonstrated the means to develop an early DFT1 diagnostic [51]. Our study found that the protein cathelicidin-3 (CATH3) was enriched in serum EVs of both devils with clinical DFT1 infection and latent devils 3–6 months before diagnosis. We hypothesised that the increase of CATH3 is likely a physiological response to the tumour, as it was independent of the tumour burden. This is a desired feature for an early cancer biomarker, as its sensitivity will be less dependent on a minimum tumour burden.

The CATH3 biomarker has several advantageous characteristics for clinical use as an early DFTD diagnostic test. Firstly, it can be measured in EVs isolated from small amounts of frozen serum, which is readily and routinely collected from wild devils; thus, a test based on the biomarker would be widely applicable. Secondly, a high diagnostic power can be achieved from a single protein rather than a multi-biomarker panel, conferring clinical practicality. Simultaneously achieving early cancer detection sensitivity and specificity levels on par with these results is uncommon with a single protein biomarker described in clinical studies [46,47,48,49]. This study reaffirms that the proteomes of extracellular vesicles in blood can be a source of reliable diagnostic biomarkers for cancer in humans and non-model species.

Implementing a preclinical biomarker into a diagnostic test to detect latent stages of DFTD would immediately aid conservation actions to preserve Tasmanian devils in the wild. Firstly, it will ensure that only healthy wild devils will be introduced into insurance populations. By pre-screening latent infected animals, the cost of maintaining devils in quarantine can be significantly reduced, which is currently required for at least 15 months (Sarah Peck, DPIPWE, personal communication). Secondly, it will improve the capacity of ongoing monitoring programs that are critical for early warning and management responses and underpin research on the epidemiology and evolutionary dynamics of this unique disease system. A three-to-six-month advantage in identifying new outbreaks will improve the capacity to respond appropriately. Finally, early detection of DFTD is critical for the effective implementation of any potential vaccination or other therapeutic intervention in the future [52].

## 8. Multi-Omics

Each omics approach has intrinsic strengths and weaknesses. Mass spectrometry-based proteomics can detect and quantify changes in the abundance of several thousand proteins but cannot (yet) approach the sampling depth of transcriptomics by mRNA-seq. Conversely, an altered gene expression manifested at the mRNA level does not necessarily predict changes at the level of protein function, particularly toward the lower range of protein abundance [53]. Workflows to support metabolomics and lipidomics are advancing rapidly but typically require greater in-house expertise to select the optimal analytical platforms for the molecular subclass(es) of interest. With the increasing availability and reduced cost of an omics analysis, it has become increasingly feasible to generate “multi-omics” datasets that facilitate data mining and biological interpretation of the results. Datasets from multi-omics experiments are complementary and may compensate for the limitations of an individual omics approach. In addition, cross-validation between orthogonal omics datasets is vastly more informative than the validation of key candidates discovered using proteomics, for example, by targeted methods such as Western blotting. Indeed, in the case of non-model species such as the Tasmanian devil, where few validated antibodies are commercially available, it may represent a more feasible approach. For example, Patchett et al. investigated the proapoptotic effects of the drug imiquimod on DFTD cells using both proteomic and transcriptomic approaches [54]. While the correlation between the two datasets was low overall, cross-validation between datasets at the functional level using bioinformatics provided detailed insights into the signalling mechanisms associated with the unfolded protein response in imiquimod-treated cells.

Multi-omics datasets can represent the flow of biological information from the genomic to proteomic levels, including post-translational modifications and the flux of metabolic end products. Although the integration of disparate multi-omics datasets may seem like a daunting challenge, it can yield a holistic understanding of a biological system. Figure 4 provides a strategic overview of how a “multi-omics” approach assisted with understanding DFTD. However, omics experiments are amenable to the application of artificial intelligence and deep learning algorithms for pattern recognition within the raw data (recently reviewed by Mann et al. [55]). The integration of multi-omics datasets can be implemented at multiple stages. It may involve steps such as dimensionality reduction to reduce the dataset’s noise and machine learning models to reduce the complexity/heterogeneity between datasets before data integration [56]. There is no doubt that in comparison with more simplistic, single omics experiments, a multi-omics analysis will become increasingly dependent on powerful computational methods and the expertise of computational biologists.

## 9. Conclusions

Research into the origin and methods to identify DFTD has led to a greater understanding of the disease process of transmissible cancers. A translation of the research techniques has led to improvements in the diagnostic tests and the potential to develop valuable techniques for preclinical or pre-diagnostic testing. Although DFTD is a rare transmissible cancer, the knowledge gained has implications for other species, including humans. The development of rationally designed immunohistochemistry panels using cross-reactive antibodies, initially developed for the studies of human cancer, may enhance veterinary/wildlife cancer identification. Furthermore, advanced molecular techniques such as RNA-seq and proteomics may provide additional insight into disease processes and origins, and the application of EV proteomics could be applied more generally for the early diagnosis of human and veterinary/wildlife tumours.

## Figures and Tables

**Figure 1 pathogens-11-00027-f001:**
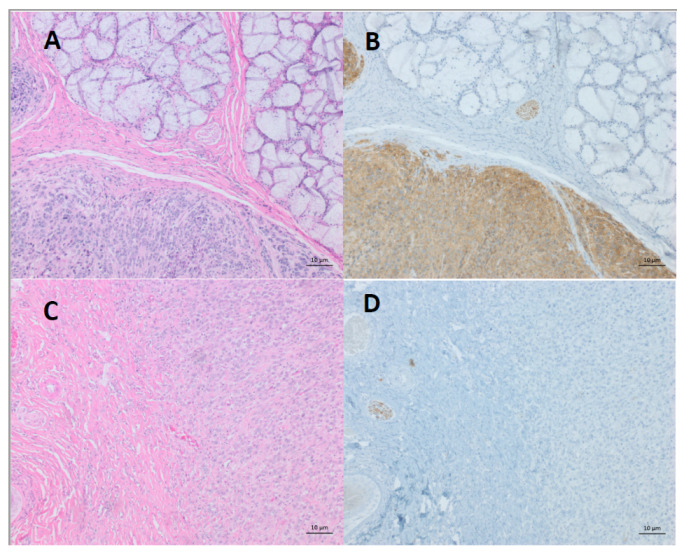
Representative histology and immunohistochemistry sections of DFT1 and DFT2. (**A**) Hematoxylin and eosin staining of DFT1 (bottom of figure) showing multinodular compact proliferation of pleomorphic round cells with a high nuclear-to-cytoplasm ratio. (**B**) Periaxin staining of DFT1 showing that the cancer cells (bottom of figure) are easily distinguishable from a peripheral nerve (upper left of figure). (**C**) Hematoxylin and eosin staining of DFT2 (right of figure), characterised by sheets of pleomorphic (amorphic to stellate and fusiform) cells arranged in a solid pattern. (**D**) Periaxin staining of DFT2 showing the staining of a peripheral nerve (middle left of figure), but the cancer cells (right of figure) are negative. Scale bars: 10 µm.

**Figure 2 pathogens-11-00027-f002:**
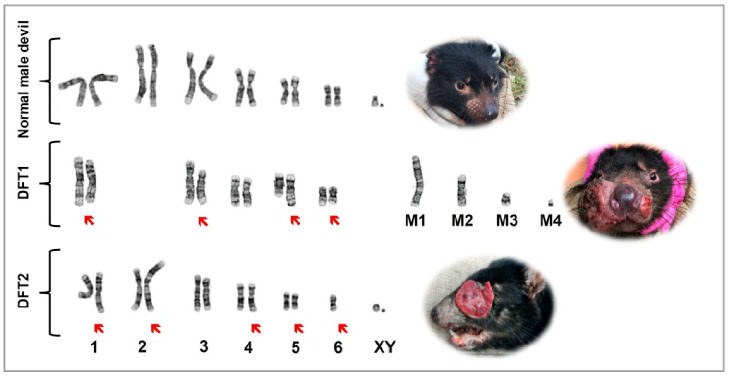
Representative karyotype of a normal male devil (top image) showing six autosomal chromosomes and X and Y chromosomes. The face of a healthy devil is shown on the right. The middle image is a representative karyotype of a DFT1 cancer showing five chromosomes and four marker chromosomes and no Y chromosome. Arrows indicate chromosomes with abnormalities. A devil with DFT1 is shown on the right. The bottom image is a representative karyotype of a DFT2 cancer showing six autosomal chromosomes and X and Y chromosomes. Arrows indicate chromosomes with abnormalities. A devil with DFT2 is shown on the right. Adapted from Reference [13].

**Figure 3 pathogens-11-00027-f003:**
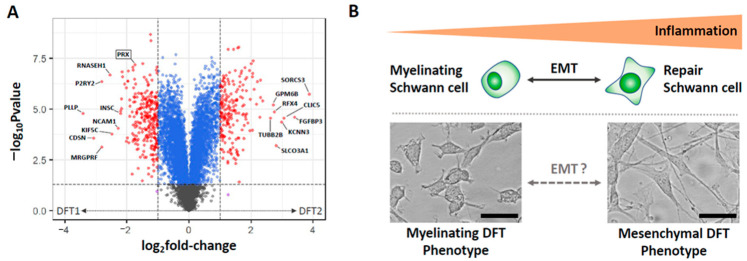
Differentiation of DFT1 and DFT2. (**A**) Volcano plot demonstrating differentially expressed proteins between the DFT1 and DFT2 cell lines that could be used in a diagnostic panel to distinguish the cancers (red points represent proteins with absolute fold-changes > 2.0, *p* < 0.05; blue points represent proteins with absolute fold-changes < 2.0, *p* < 0.05). Data was generated by Tandem Mass Tag Mass Spectrometry. Labels represent the top 8 most differentially regulated proteins for DFT1 and DFT2 and the DFT1 diagnostic marker periaxin (PRX). The plot was generated using the R package EnhancedVolcano [27]. (**B**) Proposed model of DFT differentiation. Early evidence suggests that, like Schwann cells, DFT cells may be able to transit between different states of mesenchymal activation through an epithelial-to-mesenchymal-like transition (EMT) in response to changing the immune conditions [25]. Images of representative primary DFT cell lines with a myelinating phenotype (DFT1 C5065) and mesenchymal phenotype (DFT2 JV) were captured using the EVOS M5000 imager at 20× magnification. Scale bars represent 20 µm.

**Figure 4 pathogens-11-00027-f004:**
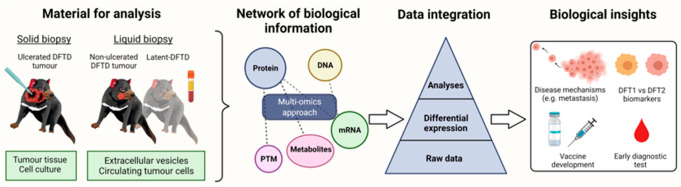
Schematic diagram of a DFTD diagnostic pipeline. Tumours samples can be collected when DFTD tumours are ulcerated in the form of a tissue biopsy, which can be used to establish primary cell cultures. In the case that tumours are not ulcerated or not present (latent DFTD), tumour biopsies are avoided. Therefore, a liquid biopsy (e.g., blood sample) will be preferred. The blood sample can be used to isolate extracellular vesicles or potentially extract circulating tumour cells. After the samples are collected, an array of “omics” techniques can be used to analyse the molecular content of the samples to find potential DFTD biomarkers (e.g., proteins, PTM = post-translation modifications, metabolites, mRNA and DNA). The information obtained by these “omics” techniques should be integrated at multiple levels, such as pattern recognition in raw data, differential expression analyses of the molecules and, finally, analysis of the data by statistical and/or bioinformatic approaches. The results of the analyses will allow the identification of potential biomarkers for early and differential DFTD diagnosis. Additionally, the results will provide insights about disease mechanisms (e.g., metastasis and immune evasion techniques), which could help the current ongoing efforts to develop an anti-DFTD vaccine.

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
