# Peer review of "Challenges of an Emerging Disease: The Evolving Approach to Diagnosing Devil Facial Tumour Disease"

_pathogens, 2021, doi:10.3390/pathogens11010027_

Round 1

Reviewer 1 Report

This manuscript is a review highlighting developments in diagnostic tools in identifying Devil Facial Tumor Disease (DFTD). The authors explore early efforts to identify and diagnose DFTD via histology and karyotyping, before detailing approaches to accurately differentiate DFTD1 and DFTD2 from each other via protein expression, phenotypic presentation, and qt-PCR. The pros and cons of solid biopsy versus liquid biopsies are remarked upon, with the authors advocating liquid biopsies as less destructive and supported by several diagnostic approaches (ie, EV-associated biomarker detection). Ultimately, the authors advocate for continuing adoption of multi-omics and the utilization of concordance between tests to accurately detect DFTD prior to the emergence of new outbreaks.

In general I found this to be a valuable review of diagnostic techniques, and the authors make a solid case for the importance of early and accurate detection of DFTD, and the need for further efforts, particularly in multiomic approaches.  Introductionary references to the 'one-health' initiative prepared me as a reader for more extensive connections to environment/human health that did not materialize. 

Specific comments:

Line 109, and line 199: Remarked that DFTD is a Schwann cell cancer, but the line of thought is not elaborated on (how this was determined and the relevance of it being Schwann cell in origin) until the line of thought is recaptured in a later section. Consider consolidating for clarity/ease of comprehension.

Line 110: Statement unclear as to what is being identified (DFTD from surrounding tissue? Schwann cell origin?).

Line 388: Prior to the final line, the implications being drawn between DFTD and human cancer diagnosis does not seem to be a strong emphasis of this review, save for a comment regarding EVs also being used as biomarkers broadly in human cancers. If this is to be the final take-away statement of the review, it may be appropriate to expand on those implications more explicitly either here or in an earlier section of the review.

Figure 4: Consider inverting the pyramid layout: elements at the base of the pyramid are interpreted as being early or foundational elements (raw data in this case), whereas it is more intuitive for elements at the point of the pyramid to be built on the preceding tiers.

Reviewer 2 Report

This article by Espejo and colleagues is a comprehensive review of the current understanding of DFTD, and provides a very useful reference for researchers interested in the field, especially as regards the specific characteristics of DFT1 and DFT2. It is concise and clearly written and the authors are to be congratulated on a very polished submission.

The only edits suggested by this reviewer concern the numbering of the references: reference 63 in Figure 3 is out of numerical sequence and the reference details are incomplete; and numbers 64 & 65 are listed in the reference list but do not have references associated with them.

Reviewer 3 Report

In the manuscript entitled “Challenges of an emerging disease; the evolving approach to 2 diagnosing Devil Facial Tumour Disease”, Espejo and co-authors reviewed the state of art of the Devil Facial Tumour Disease, reporting the current knowledge on the disease and diagnosis, and the personal experience in the field. They interestingly describe the potentiality of the use the cancer cell phenotype features in culture and the characterization of extracellular vesicles to differentiate the two kind of DFTD. The content of the manuscript is of good quality and, although relatively recently discovered, they reported extensive information.

Main considerations on limitations:

-the review is mainly based on the personal referenced experience by all the authors of the manuscript. In the text these references are reported like they are speaking of other groups es. Espejo et al described....ref 54) and ref 40, and the same for several others reference Wood, Wilson and Lyon. It is not clear which, if, come from others groups. This should be state in the text like “our experience..., we have found, ....we demonstrated”.

-The paragraph “Diagnosis of infectious diseases” is gratuitous and is not contextually addressed.  

- Lines 330-336 is more suitable for the paragraph “12. Conclusion” that is too poor. Moreover, the methodological approach comes from cancer biology, and it sound to come more from human research experience rather that from the other way round. In this context the conclusion should honestly report also the bidirectional information.

Round 2

Reviewer 3 Report

The authors have addressed all the points of concern.